# Temporal and Spatial Variability of Dryness Conditions in Kazakhstan during 1979–2021 Based on Reanalysis Data

**Irina Zheleznova [1], Daria Gushchina [1], Zhiger Meiramov [2] and Alexander Olchev [1],***

[1] Department of Meteorology and Climatology, Faculty of Geography, Lomonosov Moscow State University, GSP-1, Leninskie Gory 1, 119991 Moscow, Russia
[2] Department of Ecology and Nature Management, Kazakhstan Branch of Lomonosov Moscow State University, St. Kazhymukan, 11, Nur-Sultan 010010, Kazakhstan
* Correspondence: aoltche@yandex.ru or aoltche@gmail.com

**Abstract:** The spatial and temporal variability of dryness conditions in the territory of Kazakhstan during the period 1979–2021 was investigated using monthly and hourly ERA5 reanalysis data on air temperature and precipitation as well as various aridity indices. A large part of the territory is characterized by the air temperature increase in summer and spring, as well as precipitation reduction, especially during the summer months. It was shown that the end of the 20th century (1979–2000) and the beginning of the 21st century (2001–2021) are characterized by different trends in air temperature and precipitation. All applied indices, i.e., the Palmer Drought Severity Index (PDSI), the Keetch–Byram Drought Index (KBDI), Standardized Precipitation (SPI) and Standardized Precipitation Evapotranspiration (SPEI), showed increased dryness in most parts of the territory of Kazakhstan. KBDI indicated an increased risk of wildfires, especially in the southwestern and northwestern regions. The hottest and driest areas are situated in the regions that are simultaneously affected by rising temperatures and reduced precipitation in spring and summer. The strongest increase in aridity and fire risk in the southwest and northwest is mainly due to reduced precipitation in the summer. Minimal risks of droughts occur in the northern and central regions, where conditions in the early 21st century became even less favorable for drought formation compared to the late 20th century (increased precipitation in both spring and summer and lower summer temperatures).

**Keywords:** droughts; Kazakhstan; aridity indices; reanalysis data; temperature and precipitation variability

## 1. Introduction

Modern climate change is accompanied by a dramatic increase in global temperature, changes in precipitation patterns and growth in the frequency and severity of extreme weather events [1,2]. All these factors result in an expansion of the dryland areas [3,4] and a significant increase in the frequency of droughts in many populated regions of the world [3,5–18]. Drought is a complex natural phenomenon characterized by low soil water content caused by long-term precipitation deficits under elevated temperatures during the warm season, resulting in the depletion of soil moisture due to evapotranspiration and runoff. It can affect large areas as the weather conditions are governed by large-scale circulation processes [19]. Drought is one of the most complicated and least understood natural hazards [20,21] with widespread impacts on water resources [22], net energy budget [23], agricultural production [24], ecosystem functions [25,26], biodiversity [27,28], biogeochemical processes [29], wildfire occurrence [30,31] and local and global economies [32]. Despite numerous studies on changes in dryness and water availability, major uncertainties regarding both past and future aridity changes still remain [33–36]. Moreover, these changes are spatially not uniform and vary significantly across geographical regions. The widespread paradigm of "dry gets drier, wet gets wetter" (DDWW), commonly used as a simplification summary, was shown to be doubtful [37].

Central Asia may be classified as drought-prone and one of the most vulnerable areas to moisture deficit in the world [38–40], particularly because its economies are mainly focused on agriculture which makes this region more vulnerable to drought events [41]. Temperature increases, reduced precipitation and increased evaporation in Central Asia, documented in several studies [42–46], enhance ecosystem sensitivity to droughts because of limited water resources, low-adaptive capacity and growing population [40,47]. However, the studies of dryness conditions in Central Asia are still rather limited: Barlow et al. [48] analyzed the relationship between drought conditions and large-scale climate patterns during 1998–2001; Xu et al. [44] estimated the response of vegetation to summer droughts; Li et al. [49] described drought variation based on the Palmer Drought Severity Index (PDSI) from 1961 to 2014 and found a drying trend in Central Asia during the last decade; Guo et al. [38] investigated drought characteristics based on the Standardized Precipitation Evapotranspiration Index (SPEI) and the Run theory, demonstrating the strong heterogeneity of dryness condition trends over Central Asia. The latter implies the importance of sub-regional studies of dryness behavior given the differences in trends and spatiotemporal characteristics.

The climate changes in the territory of Kazakhstan, located in the northern part of Central Asia, have occurred somewhat faster in recent decades compared to other regions of the world situated in the same latitudinal zone: the growth rate of the air temperature from the measurements of 110 weather stations for the period from 1950 to 2020 reached 0.31 °C per decade [50]. The most notable increase in the air temperature in Kazakhstan was observed in the last two decades and was not accompanied by any significant trend in precipitation [49]. However, the surface heterogeneity and the complexity of the land surface–atmosphere interaction over Kazakhstan lead to very non-uniform distributions of more dry and more humid areas [51].

Atmospheric droughts are the most common phenomena in the territory of Kazakhstan in terms of frequency and impact on the environment, population and economy of the region [41]. Their study received particular attention in recent decades due to their high social and economic consequences for the region. Droughts occurred almost throughout the country, with varying frequency and intensity. Maximum drought frequency is observed during the growing season, and it tends to increase over time. Thus, according to Dubovyk et al. [52], there were no years without droughts for the period from 2000 to 2016. An increase in the frequency of droughts in Kazakhstan may have a number of negative effects, including soil degradation, reduction or total loss of crop yields, increased risk of wildfires, etc. [41]. Approximately 80% of atmospheric or soil droughts resulted in the significant or total destruction of agricultural crops [53]. The drought impacts in Kazakhstan are intensified by global warming and unreasonable human activities. For example, the desiccation of the Aral Sea has resulted in a more than three-fold increase in the frequency of drought occurrence in the surrounding area [47]. Estimates of projected future changes in aridity in Kazakhstan suggest an intensification of droughts under climate scenarios assuming temperature increases of 1.5 and 2 °C by the end of the 21st century (RCP2.6 and RCP4.5) [54,55]. At the same time, the projected frequency of droughts and dryland expansion are significantly higher in the RCP2.6 scenario due to projected decreases in precipitation and increased evapotranspiration, unlike RCP4.5, which implies a significant increase in precipitation in the region.

The simplest way to monitor drought conditions is to use standardized drought indices which are derived from observed measurements such as precipitation (P), soil moisture, potential evapotranspiration (PET) and groundwater levels. Some drought indices commonly used for drought assessment include the Standardized Precipitation Index (SPI), PDSI, SPEI and the Keetch–Byram Drought Index (KBDI), among others [56]. SPI and PDSI are more popular and frequently used. PDSI is the first water-budget-based index [57,58] and is calculated from precipitation, temperature and soil water content by considering both water supply and demand. However, it has some deficiencies related to the fixed timescale, strong dependence on the data calibration and no strict relationships between

drought conditions and index values [59–61]. SPI has a lot of advantages, such as simple calculation, multiple timescales, good adaptability to different climates and easy comparison between different regions and has been applied in numerous studies [62–65]. However, SPI does not consider temperature and evapotranspiration, only precipitation that may be crucial under global warming [66]. SPEI [57] is based on the water balance equation represented by the difference between precipitation and potential evapotranspiration and is widely used in the monitoring of dryness conditions on the global and regional scales [57,67–71]. Among the indices mentioned above, SPEI is more flexible and reliable for drought monitoring because it incorporates the effects of both precipitation and temperature on drought and considers the multi-scale characteristics of drought events [72,73]. KBDI is a measure of dryness conditions and is specifically designed to predict the likelihood and severity of wildfires [74]. It is able to reflect the combined effect of evapotranspiration and precipitation on the cumulative moisture deficit in the deeper and upper soil layers and hence on the flammability of surface organic matter.

Despite the high demand to study the spatial and temporal variability of droughts in Kazakhstan, as well as their consequences, there are a number of problems preventing a representative description of dryness changes in the region [52]. Most previous studies were focused on drought analysis either for only some part of the country or were conducted using data from a relatively small number of weather stations or for a limited period of time. The attention in such studies was mainly focused on agricultural regions where the main sown areas of cereal crops are located [53]. Nevertheless, these studies showed that mild droughts are very frequent in Kazakhstan, moderate droughts are less common and severe or extreme droughts are relatively rare [75,76]. The threshold values of drought intensity varied depending on the study region, the methodology applied, the time period and the number of stations used for drought analysis. It was shown that the probability of a drought recurrence for the period from 1966 to 2010 was significant for agricultural lands in western, central and northeastern Kazakhstan, approximately once every 3 years in the north and once every 4–5 years in the east. Severe droughts occurred once every 23–45 years in the north, once every 11–15 years in the east and once every 4–8 years in the western and central parts of the country [53].

Another existing problem related to accurate drought analysis in the territory of Kazakhstan is a large variety of methodologies for drought determination. The Ped index, the index of Ivanov and the hydro-thermal coefficient of Selyaninov (HTI) [51,75,77] were traditionally used in the territory of the former Soviet Union in the past [78]. Widely used in world practice, the PDSI, SPI and SPEI indices were more often applied in the analysis of arid regions of Central Asia as a whole [49,50,79], while there were a small number of incidences for their use for different regions of Kazakhstan [76,80]. Several studies to apply various vegetation [52,81] and yield indices [53] for drought analysis. The application of only one of many existing aridity indices usually results in considerable uncertainty in the estimation of dryness conditions. As a result of different approaches to drought assessment, a number of studies on the dryness conditions in Kazakhstan are contradictory. Thus, Salnikov et al. [75] estimated the aridity condition for the territory of Kazakhstan for the period 1960–2014 using the Ped index and did not reveal significant trends in aridity, while the study of Karatayev et al. [50] showed different trends of the SPEI index between 1950 and 2020. The study of Li et al. [49] for Central Asia, based on the PDSI index, indicated an increase in aridity of approximately 42% for Kazakhstan territory, and for another part of the territory (58% of the area), its decrease resulted in the weak cumulative trend of dryness reduction for the entire territory.

The latest problem in drought study in the territory of Kazakhstan is the lack of available meteorological data. In this case, reanalysis data can be a very effective tool for obtaining spatially distributed meteorological information over the entire territory of Kazakhstan, including the regions with a rare meteorological network. Modern reanalysis datasets have sufficient quality and high spatial resolution to provide detailed information

on the spatial distribution of dryness conditions throughout the study area over a long period of time.

Taking into account a very large temporal and spatial variability of dryness conditions over Kazakhstan, the high social and economic consequences of meteorological droughts for the region and a lack of generalized drought studies, the main goal of this study is to obtain new estimates of the spatial and temporal variability of dryness conditions in the territory of Kazakhstan using modern reanalysis data of the air temperature and precipitation, as well as various standardized aridity indices (SPI, SPEI, PDSI and KBDI) for the period 1979–2021.

## 2. Materials and Methods

### 2.1. Study Area

Kazakhstan is located in the central part of the Asian continent (Figure 1), and it is the ninth largest country on the planet (2.7 million km$^2$). Most of the country's territory is lowlands and plains, with mountainous areas only in the east and southeast [82]. Kazakhstan is part of the Asian Arid Zone, which is characterized by a lack of precipitation and an increased risk of droughts. The Asian Arid Zone is a geographic region that has been documented to be highly sensitive to climate change over the last 100 years and is expected to be severely affected by projected future warming [1,83].

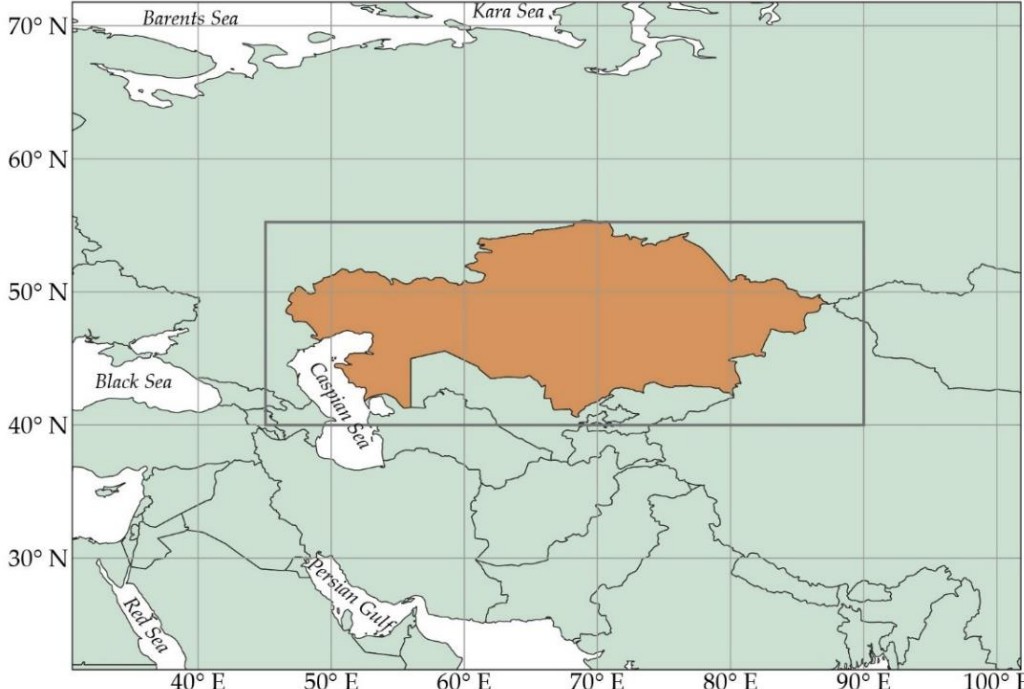

**Figure 1.** Geographical location of the study region. The gray rectangle shows the boundaries of the area selected for analysis of the spatial variability of dryness conditions.

The location of Kazakhstan in the center of the Asian continent leads to very little influence of the oceans on the country's climate. Air masses from the Indian and Pacific oceans almost do not reach this territory. The Atlantic Ocean has the largest influence on regional climate due to western transport delivering moist air masses [84].

The climate of Kazakhstan is highly continental. In the north of the country, 250–350 mm of precipitation falls annually, and in the southern regions, only 100–120 mm. The average January temperature is −15 °C, while the minimum value reaches −40 °C. The summers are fairly hot, with a maximum mean July temperature of up to 40 °C in low-lying steppes and desert steppes [84].

According to the Koppen–Geiger climate classification, the northern and eastern regions of Kazakhstan belong to cold climates with hot summer (Dfa) and warm summer (Dfb). Areas with dry steppe (BSk) and desert (BWk) climates are located in the southern and western regions [85].

### 2.2. Dryness Indices

To study the spatial and temporal variability of dryness conditions in Kazakhstan over recent decades, as a first step, six aridity indices were calculated:

- De Martonne Aridity Index (AI);
- Palmer Drought Severity Index (PDSI);
- Standardized Precipitation Index (SPI);
- Standardized Precipitation Evapotranspiration Index (SPEI);
- Keetch–Byram Drought Index (KBDI);
- Percentage of Normal Precipitation (PNI).

A preliminary analysis of the spatial patterns of selected drought indices showed that AI is not suitable for our study, as it is not intended for regions with low air temperatures, which are observed in Kazakhstan in the spring months. The spatial patterns of the SPI and PNI indices in the region under study were quite similar, so we decided to consider only the SPI index as more complex and widely used in world practice. Thus, further analysis of dryness conditions in Kazakhstan was carried out using four aridity indices: PDSI, SPI, SPEI and KBDI. The selected indices are based on a combination of different environmental parameters that allow us to take into account the influence of various factors on climate moisture conditions.

The Palmer Drought Severity Index (PDSI) is a drought index based on the soil water balance equation [86], which measures the balance between moisture demand (evapotranspiration being driven mainly by temperature) and moisture supply (precipitation) [87]. This index uses the following meteorological parameters: monthly precipitation, potential and actual evapotranspiration, the infiltration rate of water in the soil and surface runoff. The drought classification for the Palmer drought climate index is provided in Table 1 [88].

**Table 1.** Drought classification for the Palmer Drought Severity Index.

| Types of Climate | Values of PDSI |
|---|---|
| Extremely Wet | 4.00 or more |
| Very Wet | 3.00 to 3.99 |
| Moderately Wet | 2.00 to 2.99 |
| Slightly Wet | 1.00 to 1.99 |
| Incipient Wet Spell | 0.50 to 0.99 |
| Near Normal | 0.49 to $-0.49$ |
| Incipient Dry Spell | $-0.50$ to $-0.99$ |
| Mild Drought | $-1.00$ to $-1.99$ |
| Moderate Drought | $-2.00$ to $-2.99$ |
| Severe Drought | $-3.00$ to $-3.99$ |
| Extreme Drought | $-4.00$ or less |

The Standardized Precipitation Index (SPI) is based on long-term precipitation data (at least 30 years is preferred). For these data series, a probability distribution (e.g., gamma distribution) is fitted that reliably approximates long-term precipitation time series. This distribution is then converted to a normal distribution so that the mean SPI is zero [89]. Positive SPI values correspond to sufficient moisture conditions, while their negative values correspond to conditions of moisture deficiency. Drought events are indicated when the SPI values become continuously negative and reach a value of $-1$ [90]. SPI is applicable for drought monitoring and assessment of climatic conditions for time scales of one month and more (the most commonly used SPI for time intervals of 3, 6, 12, 24 and 48 months). In our study, we used SPI at the 3-month time scale (SPI$-$3).

The Standardized Precipitation Evapotranspiration Index (SPEI) is an extension of SPI, and it was developed to take into account the precipitation and potential evapotranspiration (PET) in determining droughts as key parameters influencing surface water balance [57,91]. The distribution function of the water balance is converted into a standard normal distribution. For each water balance estimate, it is possible to indicate the distribution relative to the standard deviation. If the SPEI value is less than −1, it indicates a drought. There are a number of equations for modeling PET based on available data (e.g., Thornthwaite's equation [92], the Penman–Monteith equation [93], Hargreaves' equation [94]), and the SPEI index can be calculated using any of them. In our study, we used the Hargreaves formula. It uses the difference between daily maximum and minimum air temperatures as a proxy for estimating net radiation [94]. The dryness degree categories for the SPI and SPEI indices are identical and are shown in Table 2 [95].

**Table 2.** Drought classification for the Standardized Precipitation Index (SPI) and Standardized Precipitation Evapotranspiration Index (SPEI).

| Types of Climate | Values of SPI/SPEI |
|:---:|:---:|
| Extremely wet | >2.0 |
| Very wet | +1.99 . . . +1.50 |
| Moderately wet | +1.49 . . . +1.00 |
| Near Normal | +0.99 . . . −0.99 |
| Moderately dry | −1.00 . . . −1.49 |
| Very dry | −1.50 . . . −1.99 |
| Extremely dry | <−2.0 |

The Ketch–Byram Dryness Index (KBDI) is calculated using daily data on the maximum air temperature and the rate of precipitation. The classification based on the KBDI index is shown in Table 3 [96]:

**Table 3.** Keetch–Byram drought index (*KBDI*) drought classes.

| Drought Class | KBDI Range |
|:---:|:---:|
| Very low | 0–150 |
| Low | 150–300 |
| Moderate | 300–500 |
| High | 500–700 |
| Extreme | >700 |

### 2.3. Materials

Monthly and hourly ERA5 (European Centre for Medium-Range Weather Forecasts fifth generation) reanalysis data [97] were used to calculate the aridity indices in the area of Kazakhstan ($40°−55°$ N, $45°−90°$ E) for the period 1979–2021with $0.25° \times 0.25°$ grid spacing. The data were used also to analyze the influence of various meteorological parameters on the drought intensity.

To calculate the KBDI and SPEI indices, hourly reanalysis data were used to obtain such parameters as the average, minimum and maximum daily air temperature, as well as daily precipitation amounts. To calculate the PDSI index, the mean monthly air temperatures and monthly total precipitation were also used. Only monthly precipitation data were used for SPI calculation.

The spring (March–April–May) and summer (June–July–August) periods are considered the periods when droughts in Kazakhstan are most intensive and dangerous.

The relationships between dryness indices and meteorological parameters were determined using the Pearson correlation analysis. A significance level of $p < 0.05$ was applied.

## 3. Results and Discussion

### 3.1. Spatio-Temporal Variability of Temperature and Precipitation in Kazakhstan in 1979–2021

To describe the change of dryness conditions in the area of Kazakhstan over the last few decades, the temporal variability of the air temperature and precipitation, as well as their anomalies, were first analyzed. The anomalies were calculated by removing the mean seasonal cycle for the period 1981–2010. Data analysis for the selected period of 1979–2021 showed significant interannual variability of the air temperature and precipitation in the summer and spring months (Figures 2 and 3).

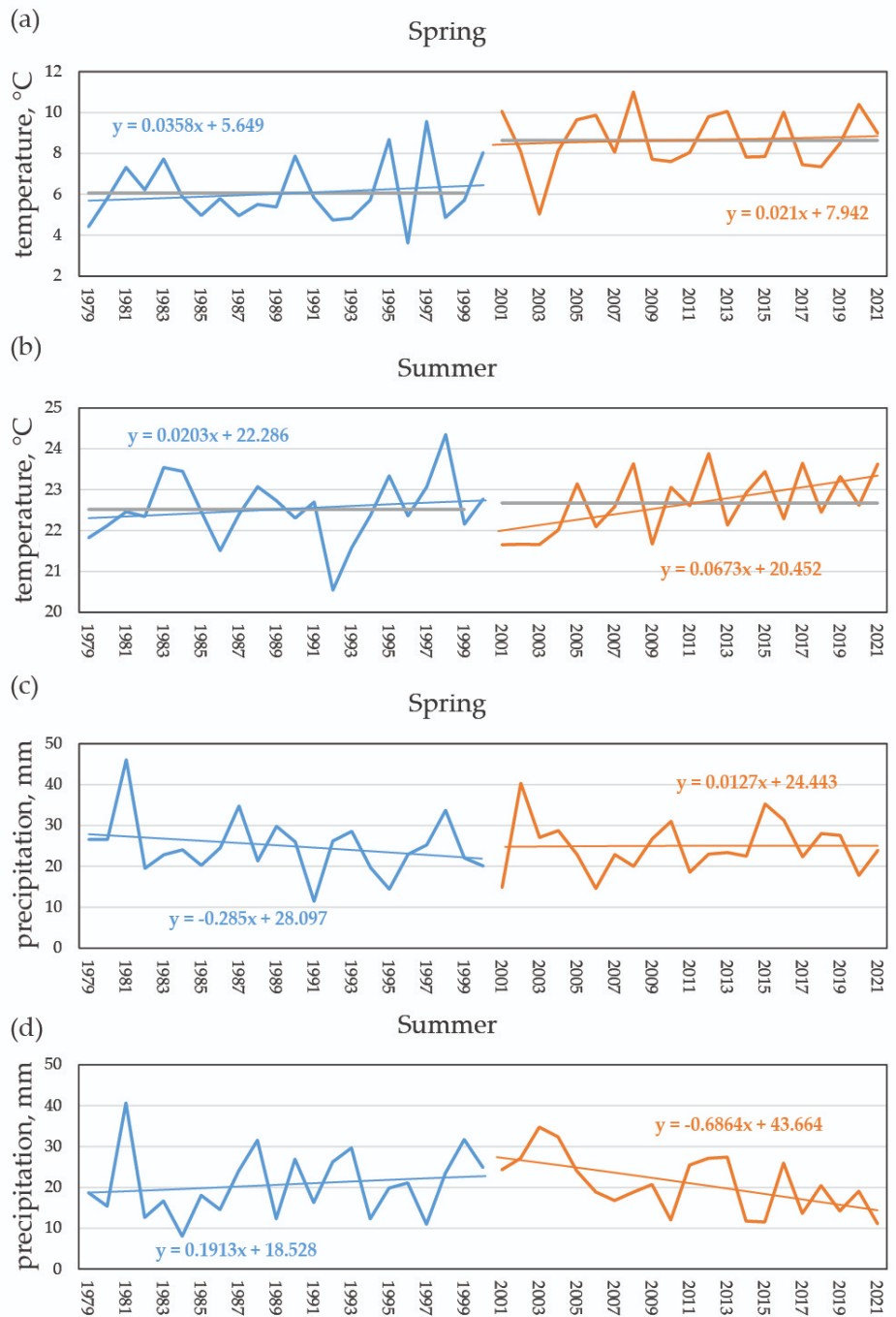

**Figure 2.** The temporal variability of the air temperatures (**a**,**b**) and monthly precipitation (**c**,**d**) averaged over spring (MAM) and summer (JJA) months in Kazakhstan. The linear trends are separately shown for the periods 1979–2000 and 2001–2021. The mean air temperatures for the selected periods are shown by gray lines.

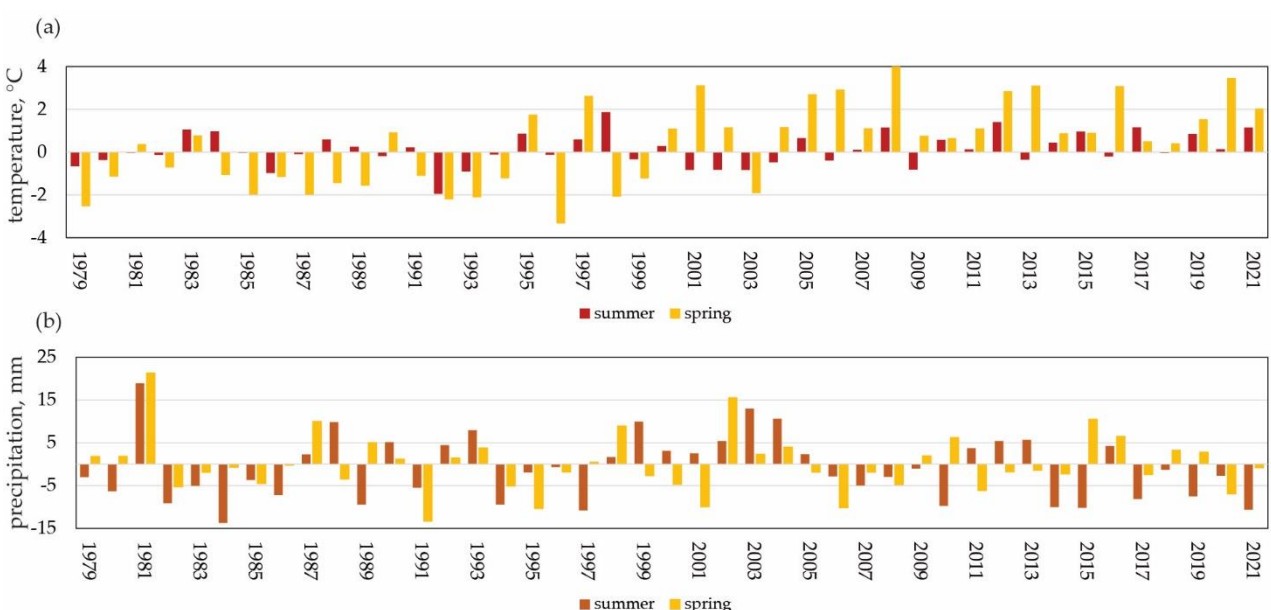

**Figure 3.** The temporal variability of annual anomalies of the air temperature, °C (**a**), and precipitation, mm (**b**), on the territory of Kazakhstan in spring (MAM) and summer (JJA) periods over 1979–2021.

Results showed that over the spring, negative temperature anomalies dominated in the late 20th century (up to −2 . . . −3 °C) and positive dominated in the early 21st (up to 3–4 °C). The temperature anomalies in the summer were less than 2 °C (Figure 3a).

The positive linear trend of the air temperature was observed during the entire time interval (1979–2021), while the differences are evident in the mean temperature value in spring and the rate of temperature growth in summer (Figure 2). For the last 20 years of the end of the 20th century, the average temperature in spring was 6.1 °C, and the summer trend was 0.4 °C/20 years, against the average spring temperature of 8.6 °C and the summer trend of 1.4 °C/20 years during the same period of the beginning of the 21st century (Figure 2a,b). The precipitation rates were also significantly varied throughout the entire study period. At the end of the 20th century, precipitation had a clear manifested negative trend in spring and a positive trend in summer. In the early 21st century, the opposite trends were documented, summer precipitation decreased while spring precipitation did not change significantly (Figure 2c,d). Taking into account the different signs of spring temperature anomalies, as well as different linear trends of temperature and precipitation in the late 20th and early 21st centuries, in further analysis, we distinguished two time intervals within the study period (1979–2021), the end of the 20th century (1979–2000) and the beginning of the 21st century (2001–2021). Similar results for Central Asia were obtained by Li et al. [49]. They showed that the rate of temperature growth in the region became noticeably stronger in the 21st century as compared to the 20th century.

Spatial patterns of the air temperature and precipitation in Kazakhstan were very heterogeneous for the spring and summer months (Figure 4). They were also differed between the late 20th and early 21st century.

Strong spring warming was observed at the beginning of the 21st century over the entire Kazakhstan area, with the highest warming rates in the central regions and the lowest in western and southeastern regions. Summer temperature changes were much weaker and characterized by both positive and negative temperature changes. Whereas the western and southern regions of Kazakhstan experienced moderate warming, the central and northern regions are characterized by temperature decreases.

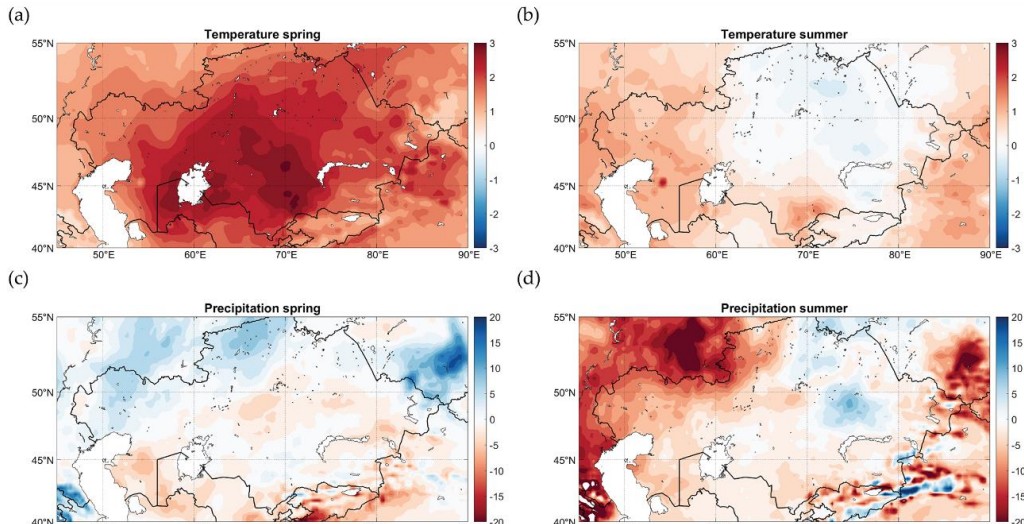

**Figure 4.** The spatial distributions of the differences between the air temperature at 2 m (upper panel **a**,**b**) and precipitation rate (bottom panel **c**,**d**) averaged over the periods 2001–2021 and 1979–2000 in the spring (left column **a**,**c**) and summer (right column **b**,**d**) seasons.

Changes in precipitation were also quite different for spring and summer (Figure 4c,d). Spring precipitation increased in the 21st century in the northern and northwestern parts of Kazakhstan and decreased in the central and southern regions. During the summer months, the western and northwestern regions become dryer, while the central and northern regions become wetter.

### 3.2. Temporal Variability of Dryness Conditions in Kazakhstan at the End of the 20th and Beginning of the 21st Centuries

Air temperature and precipitation strongly affect surface moisture conditions through changes in evapotranspiration, net radiation and surface runoff. Prolonged dry and wet spells may contribute in different ways to the occurrence of extreme events (droughts, wildfires, floods, etc.). Aridity indices are an effective tool to describe the atmospheric effect on dryness conditions of various landscape types. The maps of the difference in aridity indices between the early 21st and late 20th centuries over Kazakhstan are shown in Figures 5 and 6.

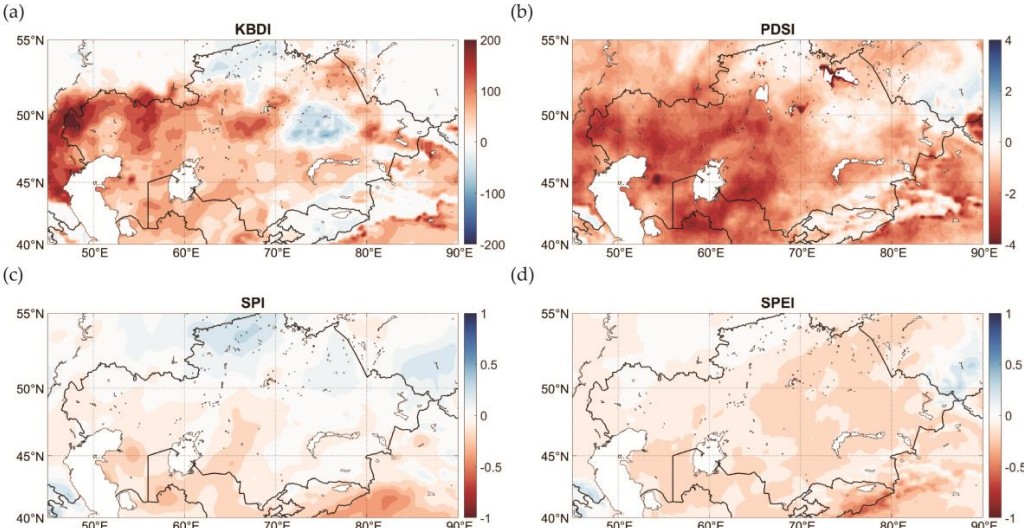

**Figure 5.** The spatial distribution of the differences between mean KBDI (**a**), PDSI (**b**), SPI (**c**) and SPEI (**d**) averaged over the periods 2001–2021 and 1979–2000 in spring (MAM).

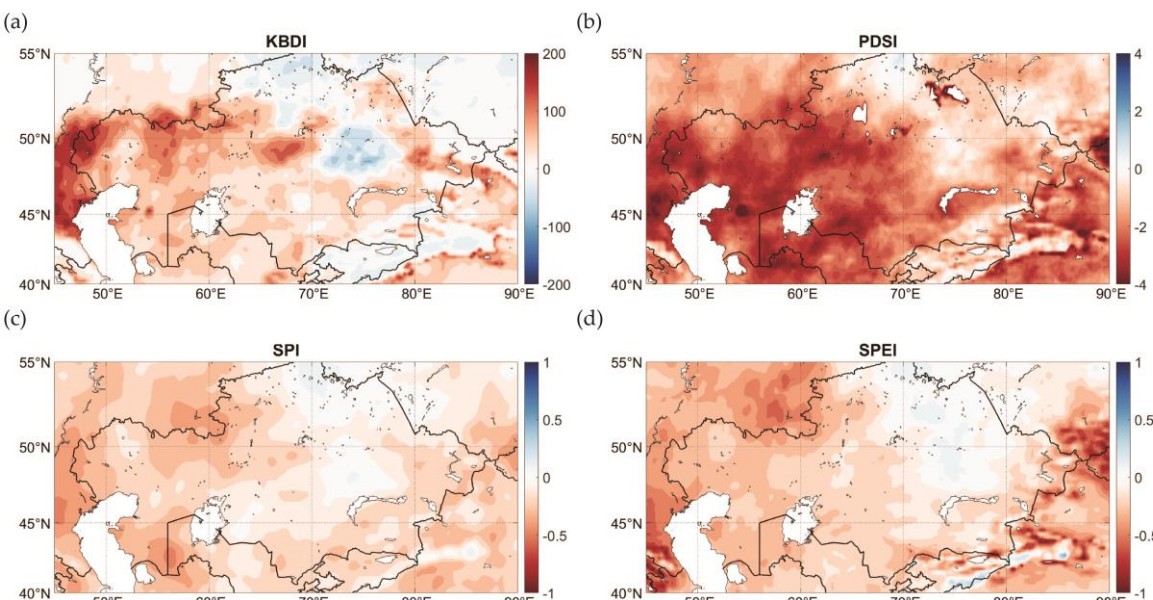

**Figure 6.** The spatial distribution of the differences between mean KBDI (**a**), PDSI (**b**), SPI (**c**) and SPEI (**d**) averaged over the periods 2001–2021 and 1979–2000 in summer (JJA).

The spatial pattern of the KBDI difference between selected time intervals (2001–2021 and 1979–2000) is characterized by strong positive differences in the western and north-western parts of the country and negative differences in its northern and central regions. It does not change significantly between spring and summer (Figures 5a and 6a). The greatest increase in fire risk in the 21st century was observed to the north of the Caspian and Aral Seas. It is noteworthy that these regions are not characterized by maximum values of KBDI that are located in the south, southwest and in the center of Kazakhstan (Supplementary materials, Figure S1).

The biggest difference between the two selected time intervals was revealed in the PDSI values. At the end of the 20th century, PDSI had slightly positive or almost zero values, which coincided with normal moisture conditions (Supplementary Materials, Figure S2a,c). In the 21st century, PDSI was significantly decreased over the entire territory of Kazakhstan, which corresponded to an increase in aridity, both in the spring and summer seasons (Supplementary Materials, Figure S2b,d). The maximum PDSI differences (up to 4 units) between the early 21st and late 20th centuries occurred in the southern, southwestern, western and most central parts of Kazakhstan (Figures 5b and 6b). The changes in the PDSI values from 0 up to −4 correspond to the transition from normal to extremely arid conditions (Table 2). An exception is the north of the country, where sufficient or episodically wet moisture conditions remained in the early 21st century.

The spatial patterns of SPI differ from the KBDI and PDSI patterns. Analysis shows that aridity does not increase in spring over Kazakhstan, while in the early 21st century, the northern regions were characterized by more humid conditions than in the late 20th century (Figure 5c). The situation changed drastically in the summer of the early 21st century when aridity increased everywhere except in the central and northern regions (Figure 6c). The maximum difference between the early 21st and late 20th centuries was observed in north-western and southwestern regions, i.e., in the areas characterized by the highest aridity.

SPEI experienced a slight decrease over the study period both in spring and summer, which coincides with aridity growth. The spatial patterns were not similar during these two seasons, with a maximum difference between two time intervals observed in the eastern part of Kazakhstan in the spring (Figure 5d) and the western part in the summer (Figure 6d). In the spring, the SPEI fell into a gradation of normal moisture conditions over Kazakhstan. In the summer, toward the 21st century, the conditions changed to moderately dry and even severely dry in the southwest and west of the country.

Therefore, analysis of temporal variability of dryness indices in Kazakhstan shows that most of Kazakhstan experienced aridity increasing in the 21st century as compared to the end of the 20th century, with the exception of northern regions. SPI, SPEI and PDSI dynamics indicate stronger aridisation in the summer, while KBDI increases more in the spring. The different behavior of various indices is noticeable. Although the indices show similar trends, i.e., an increase in wetness in the north and growth of the aridity in the west and south-west, they are not identical. The largest difference between the indices was observed in the spring months. In particular, the central regions are characterized by high fire risks (KBDI), but SPI and SPEI in these regions correspond to normal moisture conditions, and PDSI shifts to mild drought only in the 21st century. It is most likely the result of the combined effect of different factors of droughts taken into account in the different aridity indices (soil moisture, evapotranspiration, extreme temperatures, etc.).

To evaluate the changes in variability of moisture conditions between the early 21st and late 20th centuries, the difference between the standard deviations of the indices averaged over the periods 2001–2021 and 1979–2000 was also analyzed (Figure 7). The results show that the changes vary across regions, seasons and indices of aridity. The variability of KBDI increased significantly at the beginning of the 21st century as compared to the end of the 20th century, but only in the northwestern region, both in the spring and summer months (Figure 7a,b). The rest territory of Kazakhstan experienced a strong reduction in KBDI variability.

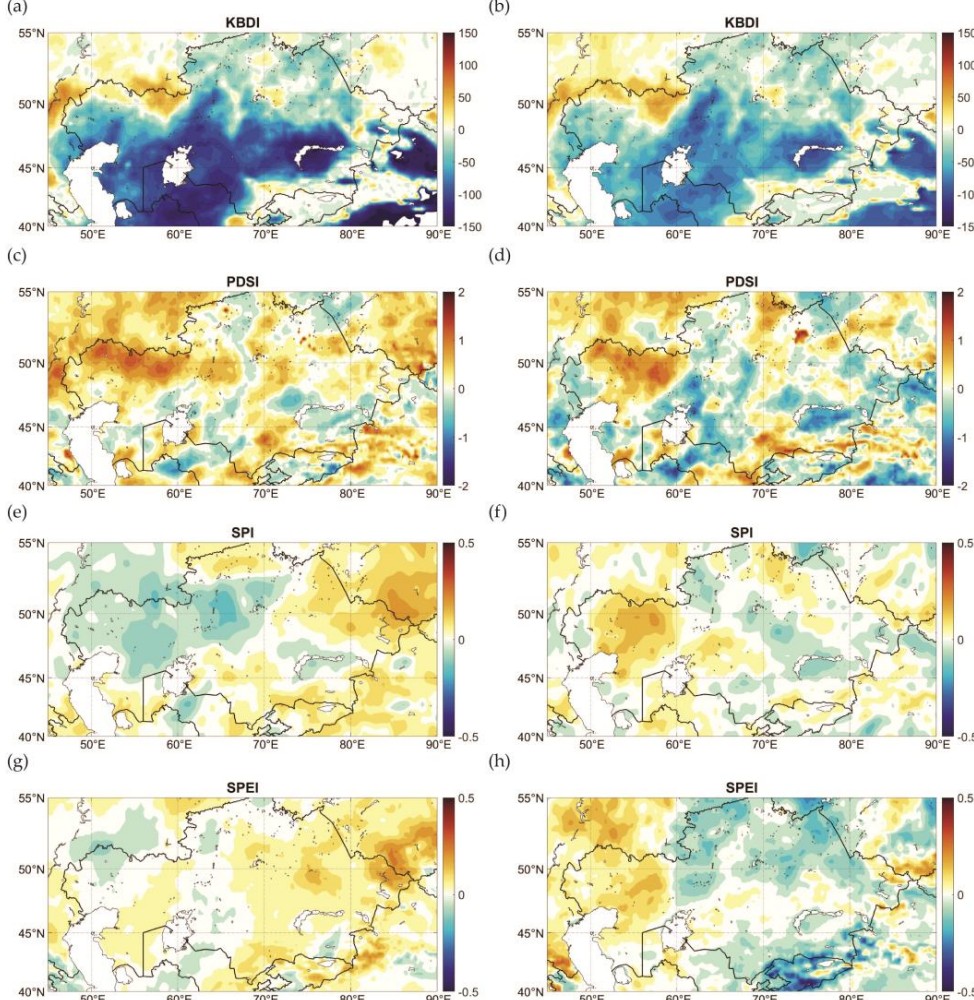

**Figure 7.** The spatial distribution of the differences in mean standard deviation of dryness indices averaged over the periods 2001–2021 and 1979–2000 in the spring (left column) and summer (right column) seasons, (**a**,**b**) KBDI, (**c**,**d**) PDSI,(**e**,**f**) SPI, (**g**,**h**) SPEI.

Changes in PDSI variability are rather heterogeneous. It increased over almost the entirety of Kazakhstan in spring, with the maximum located in the northwestern regions (Figure 7c). In summer, the increased variability was observed only in the northwestern and northern regions (Figure 7d).

The SPI variability is highly seasonal: in spring, it decreases in the western and central regions (Figure 7e), while in summer, the variability increased in the west at the beginning of the 21st century (Figure 7f).

The maximum changes in SPEI variability between the 20th and 21st centuries were detected in spring (Figure 7g). In the summer season, the variability decreased in the first decades of the 21st century over most of Kazakhstan, with the exception of the northwestern regions, where the opposite trend in the SPEI variability was detected (Figure 7h).

The combined analysis of dryness indices and their variability allowed us to reveal the regions with the highest risk of aridity growth: reduced dryness variability associated with increased aridity may imply a higher probability of droughts. In the territory of Kazakhstan, such conditions are observed in the regions situated to the west of 70°E (for KBDI) both in spring and summer (Figures 5a, 6a and 7a,b), as well as to the north of the Aral Sea (for PDSI) in summer (Figures 6b and 7d).

### 3.3. Effect of Temperature and Precipitation on Dryness Indices

To estimate the linear relationships between temperature, precipitation and dryness indices, the correlation coefficients between these characteristics were calculated (Figures 8 and 9). The results showed that air temperature is highly correlated with PDSI (Figure 8c,d) and SPEI (Figure 8g,h) in both summer and spring seasons. The lowest temperature correlation was found for KBDI in both seasons (Figure 8a,b) and for SPI in spring (Figure 8e). The spatial patterns of correlation coefficients between the dryness indices and the air temperature were quite similar for various indices in the spring season. The maximum negative temperature correlation was found for the southern and southeastern regions of Kazakhstan for all dryness indices, indicating a strong contribution of spring temperature growth to the increase in aridity and drought risks. PDSI, SPI and SPEI also emerged with a high correlation in the southwest of the country and SPEI in their central and eastern parts. The spatial pattern of temperature correlation in summer was different from the spring. The highest correlation was found in the northern and western regions, indicating a higher contribution of rising summer temperature to the occurrence of drought in these regions as compared to the spring (Figure 8b,d,f,h). In general, the correlation between temperature and dryness indices was higher in summer than in spring.

The relationships of dryness indices with precipitation are more variable compared to their dependence on the air temperature (Figure 9). The precipitation was strongly correlated with KBDI and PDSI in the southern and southwestern regions in spring (Figure 9a,c) and in the northern and eastern regions in summer (Figure 9b,d). It is noteworthy that over a large part of Kazakhstan, fire risks (KBDI) were almost not associated with spring precipitation. This may be due to the snow cover that determines the soil moisture content in spring with a weak contribution of current precipitation. SPI and SPEI were highly correlated with precipitation over the entirety of Kazakhstan in spring and summer (Figure 9 e–h). As a whole, the precipitation was more correlated with the dryness indices as compared to the air temperature.

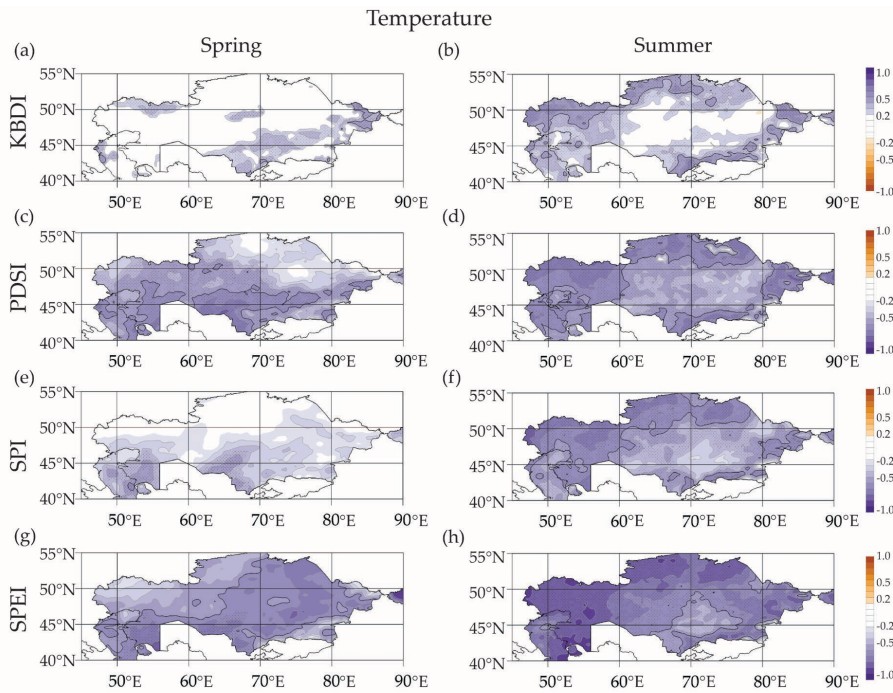

**Figure 8.** The spatial distribution of correlation coefficients between air temperature and dryness indices in the spring (left column) and summer (right column) seasons. The areas with statistically significant correlation coefficients ($p < 0.05$) are shaded. The color scale for KBDI is inverted as an increase in KBDI corresponds to the growth of aridity, while for other indices, the higher values coincide with wetter conditions. (**a**,**b**) KBDI, (**c**,**d**) PDSI, (**e**,**f**) SPI, (**g**,**h**) SPEI.

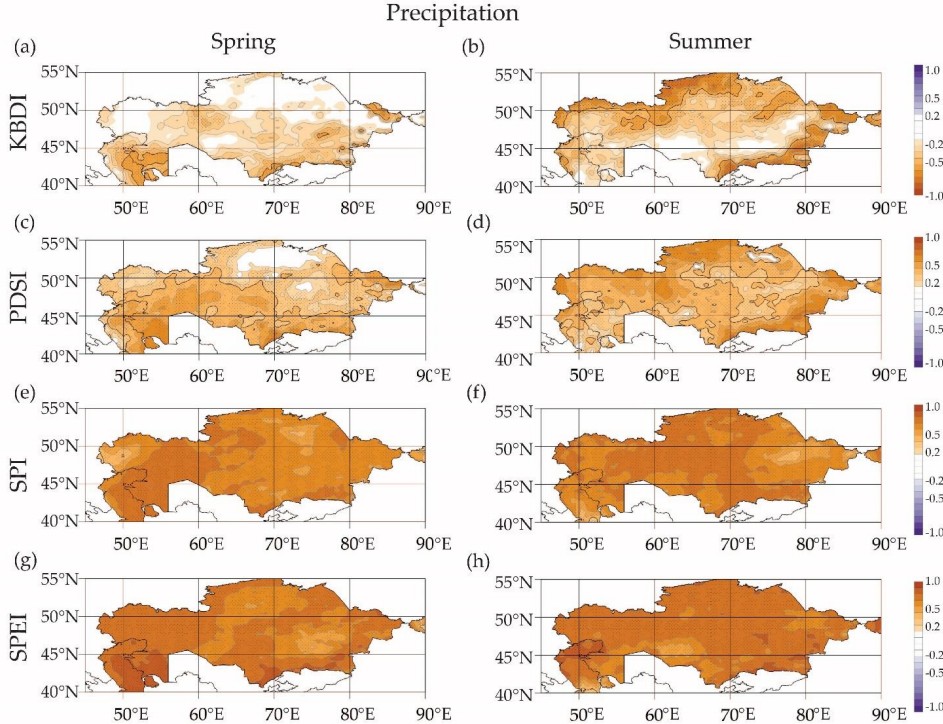

**Figure 9.** The spatial distribution of correlation coefficients between precipitation and dryness indices in spring (left column) and summer (right column). The areas with statistically significant correlation coefficients ($p < 0.05$) are shaded. The color scale for KBDI is inverted as an increase in KBDI corresponds to the growth of aridity, while for other indices, the higher values coincide with wetter conditions. (**a**,**b**) KBDI, (**c**,**d**) PDSI, (**e**,**f**) SPI, (**g**,**h**) SPEI.

## 4. Conclusions

The spatial and temporal variability of the air temperature and precipitation, as well as dryness indices in the territory of Kazakhstan, was documented for the period 1979–2021. The results showed a positive trend in air temperature over the entirety of Kazakhstan in the spring and most of the country in the summer. A large part of the territory exhibits precipitation reduction, especially during the summer months. This may favor the drought occurrence and an increase in its intensity and frequency.

Based on the trends of air temperature and precipitation in Kazakhstan, the study period was divided into two time intervals: the end of the 20th century (1979–2000) and the beginning of the 21st century (2001–2021).

To evaluate the dynamic of moisture conditions over Kazakhstan, the various dryness indices (KBDI, PDSI, SPI and SPEI) were calculated separately for the spring and summer seasons. Additionally, the differences between two selected time intervals (1979–2000 and 2001–2021) were estimated. The analysis of moisture conditions derived from various indices evidenced some differences in their spatial distribution and observed changes between the late 20th and early 21st centuries. This may be due to the contribution of different factors of droughts taken into account in used aridity indices (soil moisture, evapotranspiration, extreme temperatures, etc.) as well as to some deficiencies and limitations of these indices. It was shown that at the beginning of the 21st century most of Kazakhstan experienced dryer conditions as compared to the end of the 20th century. PDSI demonstrates a transition from normal moisture conditions to mild and moderate droughts over a large part of Kazakhstan and even to extremely dry conditions in the west and southwest of the country. The KBDI trend indicates the growth of fire risks, especially in the northwest and southwest of the country. SPI and SPEI remain in gradation of normal moisture conditions with slight positive trends (an increase in aridity), which differ between spring and summer seasons: SPI shows an increase in aridity only in summer and SPEI, in spring in the eastern regions and in summer in western regions at the beginning of the 21st century. The strongest increase in aridity occurs mostly in drought-prone regions (south, southwest and northwest). In the northern, northeastern and partially central regions, the climate conditions became wetter at the beginning of the 21st century, mainly due to increased precipitation.

The strong inter-annual variability of all dryness indices was evidenced. There was a significant increase in PDSI variability and a decrease in SPI variability in the summer at the beginning of the 21st century.

Analysis of the possible effect of temperature and precipitation on moisture conditions showed that precipitation was more correlated with all considered aridity indices compared to air temperature. In summer, the spatial patterns of correlation coefficients in Kazakhstan are similar for all indices, while the correlation of the dryness indices with temperature and precipitation is higher than in spring. In spring, the spatial patterns of correlation coefficients derived by various indices differ significantly, and the values of correlation coefficients are slightly lower. It points to more complex nonlinear processes that determine drought conditions in spring than in summer months, i.e., those may be associated with the effects of snow cover melting.

The Kazakhstan regions with the strongest increase in aridity in the 21st century (southwest, west and northwest) are affected by both positive temperature and negative precipitation trends that favor drought formation. The maximum changes are evidenced in the Actobe region, where the strongest precipitation reduction was revealed in the summer months. The northern and central regions of the country are exposed to minimum drought and fire risks, and even at the beginning of the 21st century, they were decreasing as compared to the end of the 20th century. This was as a result of the combined effect of increased spring and summer precipitation and lower summer temperature.

The obtained results for Kazakhstan are consistent with previous studies evidencing increasing aridity in several regions of Asia, particularly in China [98–100]. However, the aridity trend distribution is highly heterogeneous over various Asian regions [38,101,102].

The main mechanisms responsible for aridity trends in Kazakhstan are not yet completely known and require additional multifaceted studies. Phenomena of a wide range of time scales may be involved in the study: from decadal (phase of Pacific Decadal Oscillation), through inter-annual (Indian Dipole, El Nino Southern Oscillation), up to synoptic scale processes (regional storm tracks, synoptic variability of the Indian monsoon, etc.). The effect of all these processes is a subject of further investigation.

**Supplementary Materials:** The following supporting information can be downloaded at: https://www.mdpi.com/article/10.3390/cli10100144/s1, Figure S1: The mean spatial distributions of the KBDI index over Kazakhstan in spring (MAM, upper panel) and summer (JJA, bottom panel) for the periods 1979–2000 (left column) and 2001–2021 (right column); Figure S2: The mean spatial distributions of the PDSI index over Kazakhstan in spring (MAM, upper panel) and summer (JJA, bottom panel) for the periods 1979–2000 (left column) and 2001–2021 (right column); Figure S3: The mean spatial distributions of the SPEI index over Kazakhstan in spring (MAM, upper panel) and summer (JJA, bottom panel) for the periods 1979–2000 (left column) and 2001–2021 (right column); Figure S4: The mean spatial distributions of the SPI index over Kazakhstan in spring (MAM, upper panel) and summer (JJA, bottom panel) for the periods 1979–2000 (left column) and 2001–2021 (right column).

**Author Contributions:** Conceptualization, D.G. and I.Z.; methodology, I.Z. and D.G.; software, I.Z. and Z.M.; validation, I.Z., D.G. and Z.M.; formal analysis, I.Z. and Z.M.; investigation, I.Z. and D.G.; data curation, I.Z.; writing—original draft preparation, I.Z. and D.G.; writing—review and editing, D.G. and A.O.; visualization, I.Z.; supervision, A.O.; project administration, A.O.; funding acquisition, A.O. All authors have read and agreed to the published version of the manuscript.

**Funding:** This research was funded by the Russian Science Foundation, grant number 22-17-00073.

**Data Availability Statement:** Not applicable.

**Conflicts of Interest:** The authors declare no conflict of interest.

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
