# Peer review of "Temporal and Spatial Variability of Dryness Conditions in Kazakhstan during 1979–2021 Based on Reanalysis Data"

_climate, doi:10.3390/cli10100144_

Round 1
Reviewer 1 Report
The manuscript addresses an important scientific issue of aridisation of Kazakhstan in the last decades when climate has changed quite drastically in that region. The topic of the study is of environmental and social-economic importance. The paper represents basically a descriptive study of aridity indices changes and their links to corresponding changes of temperature and precipitation. Importantly, the authors compare a variety of aridity indices that may shed light on uncertainty and often divergent results of some previous studies. The major findings of this paper may be interesting for regional policymakers as well as for researches focused on regional aridity changes in global warming conditions. I may recommend this paper for publication after the following issues are addressed.
There are quite a few studies evaluating the moisture conditions in Kazakhstan, although there are some investigations addressing this issue for the entire Central Asia. Some of them are not cited and discussed in the manuscript. Moreover I suggest also to better highlight novelty and motivation of the this study in the Introduction.
Discussion: The results demonstrate sometimes even qualitatively different trends for the analyzed indices. It is worth putting forward a hypothesis to explain these differences for the discussion.
The language should be improved. I tried to correct some errors below. Passive voice is often misused (e.g. change “were occurred” to “occurred” in several places).
In general I suggest to shorten the text by leaving only major findings and features’ description.
Ln.32 and later in the text. Please don’t cite the whole IPCC report. Cite a specific chapter containing the implied information. One shell not go through 1000 pages to find it.
Abstract: The last sentence is hard to read. Favorable for what? Reformulate.
Ln. 40. Why obviously?
Lns.66-70. A strange notation “occurred every 11…15” and similar. Should one choose any number from the range?
Ln.109. Kazakhstan is located… not relies. I suggest shortening the whole geographical description.
Ln.177. Not “daily difference between max and min air temp”, but “difference between daily max and min air temp”.
205. No need for “Statistical analysis” subsection. Also, not need to mention Matlab software. You paid the license.
Ln.242. by Li et al.
Ln.292. differ
Ln.296. except for
Ln.311,419. As compared
Ln.315. An increase
Figure 6. It is written in the text that this figure depicts STD differences, but the caption reads “STD of dryness indices…” What is correct?
Author Response
We thank very much the reviewer for helpful and constructive comments and recommendations. The manuscript has been revised in accordance with made suggestions to produce an improved version of the article.
In attached PDF file we have provided the answers to the reviewer comments. All made changes are marked by yellow color in the revised manuscript.

Reviewer 2 Report
Comments are attached

Author Response

(The authors gave the same response as above.)
